# Highly Efficient Visible Light Active Doped ZnO Photocatalysts for the Treatment of Wastewater Contaminated with Dyes and Pathogens of Emerging Concern

**DOI:** 10.3390/nano12030486

**Published:** 2022-01-29

**Authors:** Saima Aftab, Tayyaba Shabir, Afzal Shah, Jan Nisar, Iltaf Shah, Haji Muhammad, Noor S. Shah

**Affiliations:** 1Department of Chemistry, Quaid-i-Azam University, Islamabad 45320, Pakistan; saimaaftab2013@gmail.com; 2Department of Chemistry, Women University Multan, Multan 60000, Pakistan; tayyabashabir01@gmail.com; 3National Centre of Excellence in Physical Chemistry, University of Peshawar, Peshawar 25120, Pakistan; pashkalawati@gmail.com; 4Department of Chemistry, College of Science, United Arab Emirates University, Al Ain P.O. Box 15551, United Arab Emirates; 5Department of Chemistry, Federal Urdu University of Arts, Sciences and Technology, Karachi 75300, Pakistan; mhaji21@yahoo.com; 6Department of Environmental Sciences, COMSATS University Islamabad, Vehari Campus, Vehari 61100, Pakistan; samadchemistry@gmail.com

**Keywords:** visible light active photocatalyst, doped ZnO, water contaminants, wastewater treatment, antimicrobial activity

## Abstract

Water is obligatory for sustaining life on Earth. About 71% of the Earth’s surface is covered in water. However, only one percent of the total water is drinkable. The presence of contaminants in wastewater, surface water, groundwater, and drinking water is a serious threat to human and environmental health. Their toxic effects and resistance towards conventional water treatment methods have compelled the scientific community to search for an environmentally friendly method that could efficiently degrade toxic contaminants. In this regard, visible light active photocatalysts have proved to be efficient in eliminating a wide variety of water toxins. A plethora of research activities have been carried out and significant amounts of funds are spent on the monitoring and removal of water contaminants, but relatively little attention has been paid to the degradation of persistent water pollutants. In this regard, nanoparticles of doped ZnO are preferred options owing to their low recombination rate and excellent photocatalytic and antimicrobial activity under irradiation of solar light. The current article presents the roles of these nanomaterials for wastewater treatment from pollutants of emerging concern.

## 1. Introduction

Semiconducting nanomaterials have received the utmost attention of researchers owing to their applicability in electronic devices, solar energy harvesting devices, drug delivery, water purification, pharmaceutical industries, biosensors, and ceramics [1,2]. Among nanomaterials, ZnO is of special interest for researchers thanks to its environmentally benign nature. Its band gap is 3.4 eV and it is commonly doped with elements of group I and V in an attempt to make p-type ZnO for use as a visible light active photocatalyst. The literature survey reveals that oxygen deficient sites, Zn centres, and hydrogen bond ability facilitate electron doping in ZnO [3,4,5,6,7]. Doped zinc oxides are used as visible light active photocatalysts for the degradation of emerging pollutants (EPs), which demand special attention owing to their health hazardous effects.

More than seven-hundred EPs have been reported so for. They are natural or synthetic compounds that adversely affect the human health and environment. Emerging pollutants include fungicides, pharmaceuticals, personal care products, industrial supplements, and so on. The fate and transit of developing pollutants in the environment, as well as their toxicological effects, are poorly understood [8,9]. Hence, further investigations are required for Eps’ elimination or their conversion to less harmful or nontoxic compounds. In this regard, photocatalysts are the subject of extensive investigations owing to their ability of degrading, detoxifying, or rendering the contaminants harmless. Semiconducting photocatalysts produce electron–hole pairs by the absorption of band gap matching photon. The resulting electron–hole pairs start a complex chain of reactions on the semi conductor’s surface that leads to the degradation of contaminants adsorbed on the surface of semiconductor [10,11].

Zinc oxide is photoactive and its photoactivity is further enhanced by doping. The empty octahedral sites in the hexagonal closed packed structure of zinc oxide generate interstitial gaps that are occupied by other atoms or defects. These imperfections add to zinc oxide’s photocatalytic activity [12]. A variety of methods such as sol–gel, hydrothermal, spray pyrolysis, microwave-assisted procedure, chemical vapour deposition, ultrasonic condition, and precipitation approaches are used for the synthesis of zinc oxide. Compared with chemical methods, biological methods of its synthesis are gaining more popularity because they are simpler, cheaper, and safer [13,14].

Doping of ZnO is done to lower its band gap so that it can absorb visible light, which constitutes 40–45% of sunlight. Doping also alters the electronic and optical characteristics of photocatalysts [15]. Hence, doped zinc oxide is used in various fields, from tyres to clay, medicines to agriculture, and paint to chemicals [16,17]. The resilience of ZnO to high energy radiations implies its entrenchment for space utilization, the tendency of ZnO to etch in acidic or basic solutions implying its suitability for manufacturing smaller devices, and its sensitivity to chemical species are additional appealing properties [18,19]. Moreover, ZnO has been proven as an effective photocatalyst for the degradation of various pollutants [20,21,22]. Its photocatalytic activity is further increased by quantum confinement effect by tuning the electron energy band gap. Although a few reviews on visible light active photocatalysts are available, a review of recent reports is always needed to update the readers about this research field. Moreover, in comparison with previous review articles focused on various pollutants, the present review is particularly dedicated to the removal of highly hazardous pollutants of emerging concern using doped zinc oxides as potential photocatalysts. To the best of our knowledge, this is the first report on doped zinc oxides nanomaterials that target the elimination of extremely toxic dyes and pathogens from water, and thus hold promise for protecting ecological and human health.

## 2. Structure and Properties of ZnO

Crystallized ZnO in wurtzite hexagonal lattice form possesses oxygen ions in the tightest hexagonal packing. The tetrahedral locations are occupied by zinc ions (Figure 1). On the hexagonal side, or c axis, the spacing between close neighbours is slightly smaller than the other three. Although the interaction is polar according to the c axis, there is a homopolar component of binding between zinc and oxygen ions [23].

## 3. Visible Light Active (VLA) ZnO Photocatalysts

The development of VLA photocatalysts to challenge water contamination is attracting the attention of researchers, because visible light makes up a significant fraction of the solar spectrum. Accordingly, many new photocatalytic materials have been recently developed to disinfect water. ZnO NPs is one of these materials, thanks to its excellent quantum efficiency and electrical structure that prevents the chances of recombination of electrons/charge in the electrolyte. Accordingly, many new doped ZnO-based photocatalytic materials have been developed for disinfecting water. The band width of ZnO is altered with defects like doping with non-metals and metals [24]. Figure 2 indicates the general characteristics of dopants to increase the photoactivity of ZnO nanoparticles in the presence of visible-light.

The objective of reducing the band gap of ZnO is to make the device work in the visible light range. There can be a decrease in the band width and activation under visible light by producing new energy levels within the band gap. By combining components, the band gap can be narrowed and, within the band gap, structural inadequacies lead to new levels of energy [25]. This has been depicted in Figure 3 by a simplified illustration where donor levels are introduced owing to the hydrogenation of zinc oxide microstructures within the gap between bands.

ZnO NPs doped with non-metals have emerged as excellent materials owing to greater photocatalytic activity under solar light and a low recombination rate [26]. By incorporating additional components such as S, F, and N, it is possible to minimize the band gap in ZnO [27,28]. Figure 4a shows the valence band (VB) of pure zinc oxide, which generally consists of O 2s, O 2p, and Zn 3p states, whereas O 2p, Zn 4s, and Zn 3s states dominate the conduction band. The fermi levels set to zero are shown by the dotted line. Figure 4b demonstrates the direct band gap of zinc oxide, where the CB minima and VB maxima are both located at the same symmetric gamma G point. The CB minima and VB maxima are placed at the equivalent symmetric gamma G point, which show direct bandgap properties of ZnO. Furthermore, these non-metallic dopants of C and P have little impact on the lattice constant (6.58 A°) of ZnO, which changes to 6.60 in the case of C doping and 6.62 in the case of P doping. The conduction band changes to a lower energy in comparison with pure ZnO [29]. Incorporation of B and F in ZnO has been reported to increase the electrical properties [30].

Nitrogen is considered as the most effective dopant for p-type ZnO. However, obtaining consistent and stable p-type doping poses significant difficulties. For instance, there is no way to compensate the indigenous handicap caused by the usage of n-type doping. Research on doping N and N_2_ in ZnO has revealed that N_2_ at the Zn site acts as a shallow recipient, and at the O-site, it acts as a donor, while N acts as a deep recipient [31,32,33,34]. A number of methods such as molecular beam epitaxy, active sputtering, chemical fumes, and ZnO ammonolysis at low temperatures are used for producing nitrogen-doped ZnO. Using the proper nitrogen supply and promoting nitrogen introduction into oxide using a solution combustion method is a simple, yet promising method of nitrogen fixation in oxide [35]. N-doping in ZnO causes narrowing of the band gap in ZnO and increases the absorption of visible light [36,37]. The photocatalytic activity of ZnO towards the breakdown of organic dyes is increased by enhanced VL absorption, which is ascribed to the formation of localized states of N 2p in the bandgap [38].

## 4. Use of ZnO for the Breakdown of Dyes

Zinc oxide is extensively used for the breakdown of potentially harmful water pollutants. The majority of commonly used water treatment methods such as chemical treatment, adsorption, and membrane filtration are useful, but inadequate for complete removal of pollutants from wastewater [39]. Additionally, these approaches produce by-products such as sludge as solid waste and toxic gases that need further treatment. Therefore, for effective water clean-up, visible light active zinc-oxide-based photocatalysis is receiving special attention. Photocatalysis uses photoexcited charge carriers for the breakdown of organic contaminants. The photoactivity of ZnO is utilized for destroying such water toxins [40].

Shinde et al. [41] investigated two methods of dye degradation. In one method, the transport of electron into CB of photocatalyst occurs by photon having energy equal to or greater than the band gap of nanoparticles. The excited electrons result in the formation of holes in valence band. The generated electrons and holes leads to free radical formation [42]. The oxidation of pigments is triggered by the holes present in the valence band.
h^+^_VB_ + dye → dye * → dye degradation

The second method involves dye sensitization, where dye on the water surface absorbs visible radiation. The absorbed photon excites electrons of dye molecules from HOMO to LUMO. The electron present in LUMO shifts to the conduction band of ZnO. On reaction with O_2_, this electron yields O_2_^−^, which causes the breakdown of pollutants present in water. The schematics of both methods can be seen in Figure 5.

T. Bora et al. [43] reported the potential antibacterial activity of ZnO nanorods and their photocatalytic activity towards the breakdown of methylene blue and phenol (Figure 6). The rate constant for photocatalytic breakdown of methylene blue dye and phenol was found to be 0.032 and 0.094 min^−1^, respectively, at 250 °C. Similarly, ZnO doped with Ag degrades organic dyes by the prevention of hole and electron recombination, which enhances the photocatalytic activity, as expected [44]. The degradation efficiency of ZnO doped with silver was found to be 99.64% against rhodamine B.

Kumar et al. [45] also discovered that the photocatalytic activity of ZnO is increased on doping with Ag. Nanoparticles of Ag-ZnO developed with 0.02, 0.04, and 0.06 percent of Ag showed 3.03 eV, 3.01 eV, and 2.96 eV band gaps, respectively. As discussed earlier, doping lowers the energy required to shift an electron from valence band to conduction band [46]. William et al. [47] used Ag-doped ZnO thin film for the degradation of methylene blue. The results of their experiments revealed that silver-doped ZnO degrades dye more effectively (45.1%) than a pure ZnO (2.7%) film under irradiation of VL. In the same area of study, Sabry et al. [48] carried out some modification in Ag-doped ZnO nanostructure by introducing stearic acid for the breakdown of methylene blue and achieved 93% breakdown efficiency after 80 min exposure to visible light. Liu et al. [49] also used doped Ag-ZnO and found it to be effective for the degradation of Congo Red. Figure 7 presents the optical properties of doped ZnO by photoluminescence and UV/Vis absorption spectroscopy. Using the W Xe lamp, Congo Red, Methyl Orange, and Methylene Blue were photocatalytically broken down. A 0.25 M Ag-doped ZnO showed maximum photocatalytic activity of 91.9% in 120 min under a solar light simulator.

Saffari et al. [50] prepared pure ZnO and P-containing ZnO for Rhodamine B degradation. Using a halogen lamp, visible light in the wavelength range of 375–1000 nm was employed. A lowering in the electrons and holes recombination rate in the sample containing 1.8% of phosphorus resulted in complete breakdown of RhB dye in 180 min. Kuriakose et al. [51] applied Cu-doped ZnO for the degradation of methyl orange and methylene blue dyes. Furthermore, 5% Cu doping in ZnO led to 92% and 80% degradation of methylene blue and methyl orange, respectively, in 30 min. Vaiano et al. [52] used ZnO doped with Cu for photocatalytic oxidation of As (III) to As (V). Here, 52% photocatalysis was observed after VL exposure of 180 min for ZnO doped with 1.08% of Cu. Similarly, Kamlesh et al. [53] employed Cu-doped ZnO nanoparticles for methyl green degradation and, under irradiation of VL, noticed a 3.5-fold enhancement in photocatalytic activity against methyl green as compared with undoped ZnO.

Vinodkumar et al. [54] studied the impact of Mg doping on the photocatalytic performance of ZnO. Compared with ZnO, 0.1% MgZnO showed a twofold higher photocatalytic efficiency. Adam et al. [55] also synthesized Mg-doped ZnO nanoparticles and investigated their role for the degradation of methylene blue. ZnO on doping with 0, 3, 5, and 7% of Mg resulted in approximately 55, 65, 77, and 96% decolouration of MB dye, respectively. The improved photocatalytic performance by Mg-doped ZnO can be attributed to the role of Mg in the ZnO lattice that enhances the hydroxyl ions’ absorption at the nanoparticle surface and works as trap sites to increase the photodegradation as well as to lower the electron–hole pair recombination. Adeel et al. [56] investigated ZnO doped with Co to remove methyl orange from wastewater. Figure 8 shows that ZnO doped with 10% Co causes 93% methyl orange degradation. The authors ascribed the increase in photocatalysis to the prevention of electron–hole recombination.

Hernandez et al. [57] compared the photocatalytic efficiency of undoped and Eu doped ZnO. A 99.3% breakdown of methylene blue dye was achieved with doped nanoparticles. Similary, Tb- and Eu-doped ZnO NPs showed 100% photodegradation of methylene blue and facilitated reduction of CO_2_ and production of H_2_ [58]. Petronela et al. [59] synthesized Ni-Co codoped ZnO NPs to check their efficiency towards the removal of Rhodamine B dye from contaminated water in the presence of visible light, which was found to be 42% for 0.69% of both of the dopants. Similarly, Shanmugam et al. [60] probed the photocatalysis of ZnO/Cu/Sn nanoparticles and found these to have much better photocatalytic activity towards methylene blue, which was completely degraded in 180 min under visible light irradiation.

Qin et al. [61] observed the ability of ZnO-graphene nanocomposites for the removal of methylene blue from wastewater. High absorptivity of the dye and charge separation process led to enhanced photocatalytic degradation efficiency of the nanocomposites. Ruhullah et al. [62] prepared Mn-doped ZnO and found these to be effective for the breakdown of basic aniline dye and methylene blue dye. In another study, carbon nanotubes supported Mn-doped ZnO NPs efficiently photodegraded >95% malachite green in water under irradation of VL [63]. Similarly, Labhne et al. reported photodegradation of rhodamine B (99% in 140 min) and congo red (100% in 160 min) using Mn-doped ZnO supported with reduced graphene [64]. Maline Ghosh et al. [65] used a bimetallic photocatalyst, which resulted in 90% breakdown of caffeine with a rate constant of 0.024 min^−1^. Subhan et al. [66] prepared a trimetallic oxide nanocomposite ZnO.CuO.CeO_2_, which exhibited 97% photocatalytic dye degradation activity. ZnO nanorods doped with Nd and Gd were also reported for methylene blue degradation under illumination of VL. A 1.5% Nd-Gd codoped ZnO resulted in 93% photocatalytic breakdown of MB in 180 min [67]. The photocatalytic efficiency of other doped zinc oxides for the degradation of various dyes can be seen in Table 1 [68,69,70,71,72,73,74,75,76,77]. Thus, it can be concluded that doping increases the photocatalytic characteristics of ZnO nanostructures.

## 5. Water Disinfection with Visible Light Active ZnO-Based Photocatalysts

Booming industrialization has led to water quality deterioration. Chlorination and ozonization methods are frequently used for disinfecting water. However, these methods are not free from limitations; for example, various carcinogenic by-products are produced during chlorination and ozonization [78,79,80]. Photocatalysts hold great promise for effectively treating contaminated water. Among the photocatalysts, ZnO has received enormous attention as a disinfectant thanks to its stable nature under severe processing environments [81]. Moreover, it does not cause secondary pollution. Its phase composition, particle size distribution, defective surface, and specific surface area play their role in antibacterial/antimicrobial activity. ZnO nanoparticles adopt different ways for their antimicrobial activity, such as release of oxygen species and Zn^2+^, which can kill microorganisms by damaging their DNA and cell membrane. ZnO also exerts antimicrobial activity when it comes in direct contact with the cell membrane of microbes [81,82,83,84]. Gancheva and colleagues [85] utilized zinc oxide powder for disinfecting water. The azo dye of Malachite Green was removed from water in 180 min using ZnO powder in the presence of visible light during the disinfection procedure. Mahamuni et al. [86] found that ZnO particles had varying degrees of antibiofilm and antibacterial action against Gram-negative and Gram-positive Staphylococcus and Proteus vulagaris. The antibacterial and antibiofilm activity of ZnO nanoparticles was found to vary inversely with particle size. At a concentration of about 50 μg/mL, it inhibited 32.67 percent of Staphylococcus aureus and 22.38 percent of Proteus vulgaris, respectively. The greatest biofilm resistance against Staphylococcus aureus and Proteus was 67.3 percent and 58.18 percent, respectively, for 250 μg/mL concentration of ZnO. In one particular study, Haque et al. [87] reported that ZnO prepared by the biological method degraded methylene blue dye up to 80% in 20 min, compared with 68 percent by ZnO produced by the sol–gel approach. So, according to this study, ZnO prepared by the biosynthetic approach outperforms ZnO prepared via the sol–gel method in terms of water disinfection. Working in this area, Ogunyemi, et al. [88] found the maximum antibacterial activity of ZnO NPs produced via using olive leaves compared with chamomile flower and red tomato fruit (Figure 9). Olea europaea ZnONPs had the maximum inhibitory zone of 2.2 cm at 16.0 μg/mL. The authors attributed this excellent activity to the small crystallite size of ZnO NPs.

Similarly, J. Suresh et al. [89] also synthesized zinc oxide NPs in combination with *Costus Pictus D*. leaf extracts. Biosynthesized zinc oxide nanoparticles showed greater antimicrobial activity against fungal and bacterial species. The zone of inhibition for *B. subtilis*, *S. aureus*, *S. paratyphi*, and *E. coli* was found to be 17, 10, 12, and 10 mm, respectively. Inhibition and cell survival were found to increase with the increasing concentration of ZnO nanoparticles. Working in this area, Panchal et al. [90] mixed seed extract from Ocimum tenuiflorum to Ag/ZnO NPs and observed improved antimicrobial activity in comparison with pure NPs and Ag/ZnO. In 15 min, bacteria with a density of 1 × 10^8^ cfu were killed by 1.0% Ag/ZnO nanocomposite.

Weiwei He et al. studied the antibacterial activity of Au/ZnO hybrid nanostructures [91] and found that deposition of gold nanoparticles in the molar ratio of 0.2% on ZnO surface significantly increased the antibacterial activity of ZnO. ZnO/Au nanostructures were found to be nearly three times more efficient than pure ZnO nanoparticles at killing *E. coli*. Similarly, doping ZnO with other nanoparticles such as Ag, Sn TiO_2_, and Cu also show improved properties. For instance, Shanmugam et al. [60] developed an Sn/Cu-doped ZnO photocatalyst and found that it completely removes methylene blue in 180 min under visible light. They found an increase in the photodegradation rate of doped ZnO in contrast to undoped ZnO by a factor of 2.6, and the photocatalyst showed good antibacterial activity against the pathogens *E. coli* and *S. aureus* (G+).

Qamar et al. [92] discovered the antimicrobial properties of hybrid g-C_3_N_4_/Cr-ZnO nanocomposite. These nanocomposites were found to cause 93% removal of methylene blue in 90 min. The enhancement in photocatalysis of hybrid composite was due to good separation of electron and hole pairs and absorption. Good antibacterial activity of composites was observed against Gram-positive (*Bacillus subtilis*, *Staphylococcus aureus*, and *Streptococcus salivarius)* and Gram-negative (*Escherichia coli*) bacteria. The inhibition zone was found to be approximately 18, 19, 17, and 15 mm for *B. subtilis*, *S. aureus*, *S. salivarius,* and *E. coli*, respectively. Munawar et al. [93] synthesized heterostructured nanocomposite ZnO-Er_2_O_3_-Yb_2_O_3_, which demonstrated enhanced antibacterial activity against *E. coli* (*Esherichia coli*). The Fe/ZnO nanoparticles prepared by Das et al. photocatalytically disinfected multidrug-resistant *E. Coli* in true samples from a municipal tap, river, and pond. Fe/ZnO nanoparticles were found to disinfect *E. Coli* completely in 90 min [94]. Li-doped ZnO NPs proved to be promising against *E. faecalis*, *S. tyjimurium*, and *Esherichia coli* [95]. Darwish and his group [96] reported the potential of Sm/Ag/ZnO/cuttlefish bone nanorods in destroying *S. mansoni* worms. The percentage inhibition was documented to be 20 and 47% for *P. aeruginosa* and *S. aureua*, respectively. These literature findings depict that doped ZnO NPs endowed with photocatalytic and antimicrobial activity are ideal nanomaterials for water treatment and disinfection applications.

## 6. Conclusions

The presence of water toxins, especially dyes and pathogens, is challenging scientists to search for highly effective, cost affordable, and environmentally friendly methods for water purification. This driving force has led to remarkable scientific research in wastewater treatment. One of the environmentally friendly technologies for water treatment is the use of visible light active photocatalysts. In this regard, ZnO has been given special attention as it is nontoxic and its band gap is switched by earth abundant elements of group I and V, making doped zinc oxides cost-effective and suitable for working under visible light, which is a major (45%) portion of sunlight. The results presented in this document reveal the competency of doped zinc oxide for disinfecting water under visible light irradiation. Doping reduces the chances of electron and hole pairs’ recombination and, consequently, their availability initiates free-radical-based degradation of water pollutants. The results reveal that the photocatalytic performance of doped ZnO depends on various factors such as the concentration of the dopant, their energy states within the ZnO lattice, localized states of dopants in the bandgap of ZnO, dopants sites’ distribution, electron hole pair recombination, and the intensity of incident light. ZnO nanomaterials have the ability to remove microbes and organic dyes from wastewater. The commercial application of doped ZnO nanoparticles can be anticipated in the near future owing to their nontoxic, low-cost, improved photocatalytic, and antimicrobial activity. The presented work targets the elimination of extremely toxic dyes and pathogens with the potential to remove them from water, thus protecting ecological and human health. However, highly efficient photocatalytic degradation and antimicrobial activity still demand more research to design more effective doped ZnO NPs and ZnO-based nanocomposites. Future endeavors will be focused on beating the challenge of synthesizing doped ZnOs with a suitable bandgap. The selection of materials remains a vital consideration for achieving this future targeted objective.

## Figures and Tables

**Figure 1 nanomaterials-12-00486-f001:**
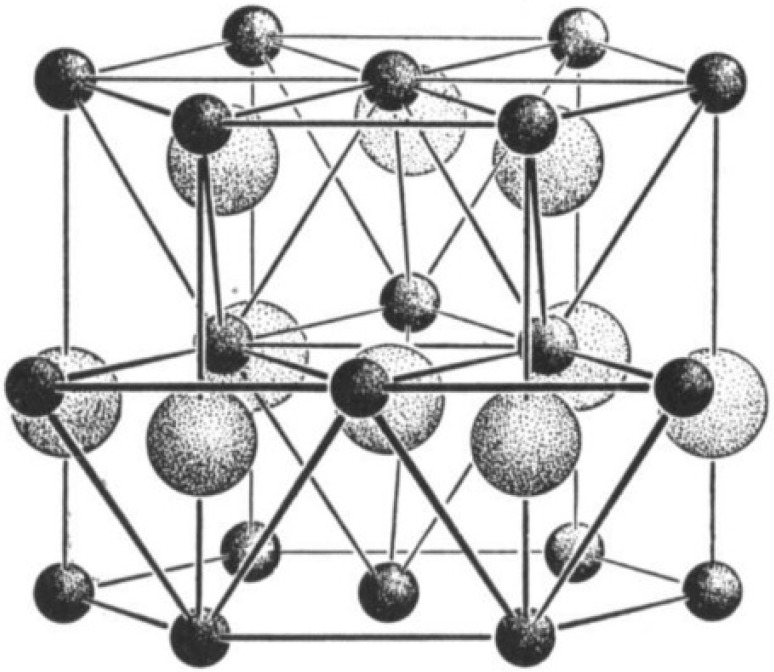
The wurtzite lattice of zinc oxide. Adapted with permission from [23]. Copyright 1959, Elsevier.

**Figure 2 nanomaterials-12-00486-f002:**
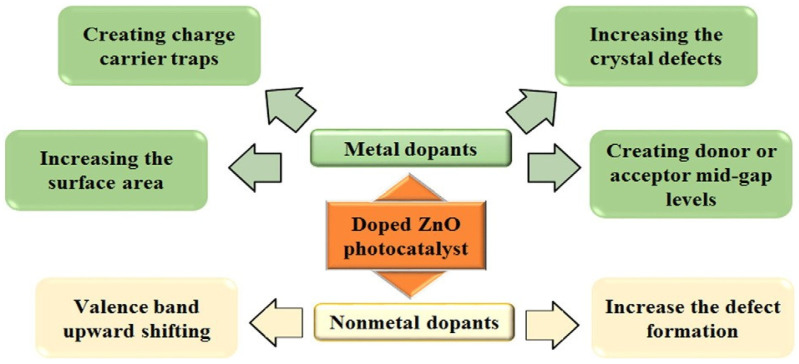
General properties of dopants for increasing the visible-light photocatalytics of ZnO. Adapted with permission from [24]. Copyright 2016, Elsevier.

**Figure 3 nanomaterials-12-00486-f003:**
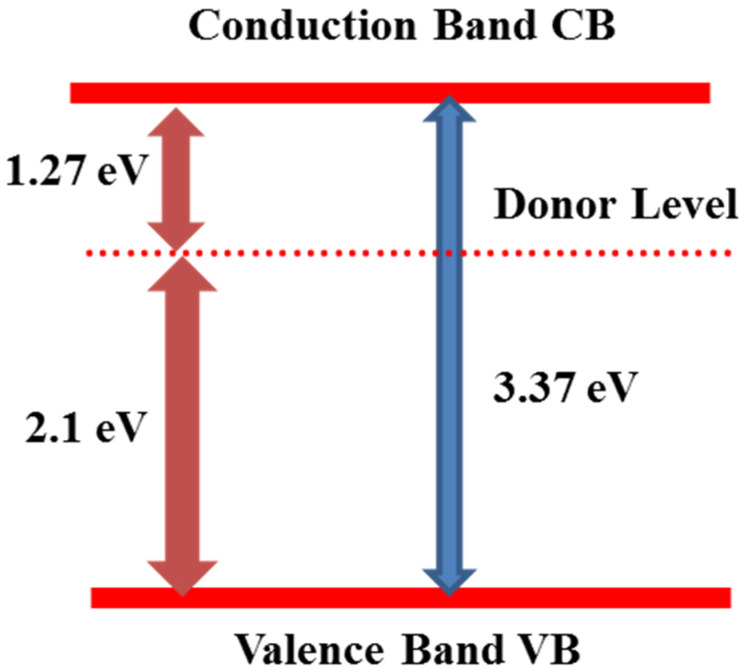
Diagram indicating the introduction of new energy levels inside the band gap of ZnO. Adapted with permission from [25]. Copyright 2019, KeAi Publishing.

**Figure 4 nanomaterials-12-00486-f004:**
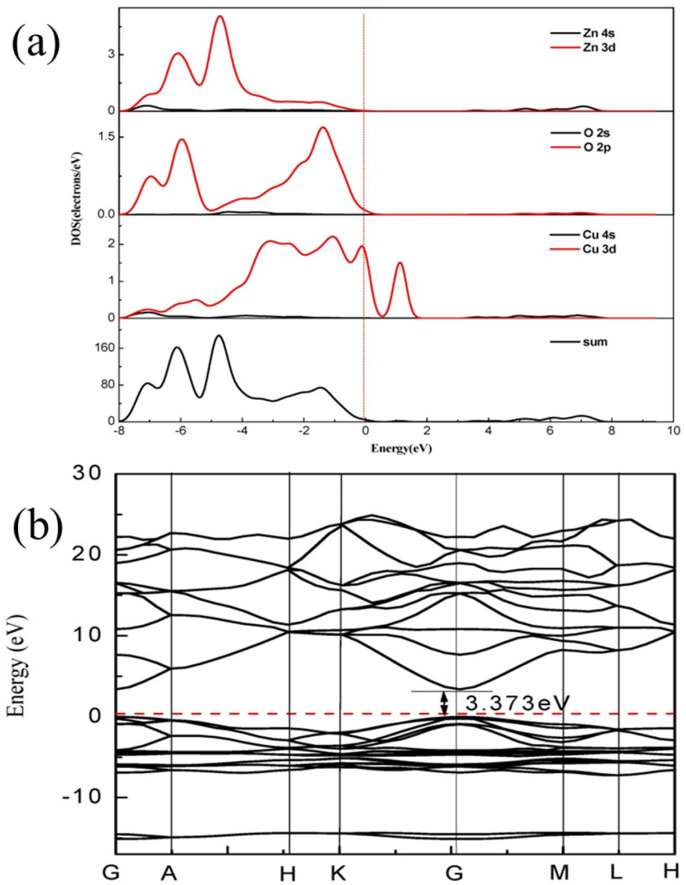
(**a**) Total density of state and (**b**) electronic band structure of pure ZnO reproduced from [29].

**Figure 5 nanomaterials-12-00486-f005:**
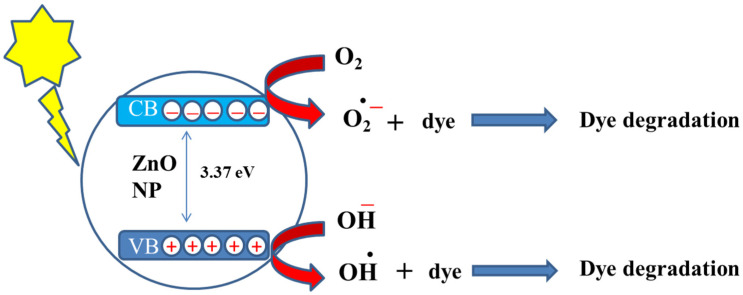
Schematics of the photoexcitation of the dye followed by its photocatalytic degradation under solar irradiation.

**Figure 6 nanomaterials-12-00486-f006:**
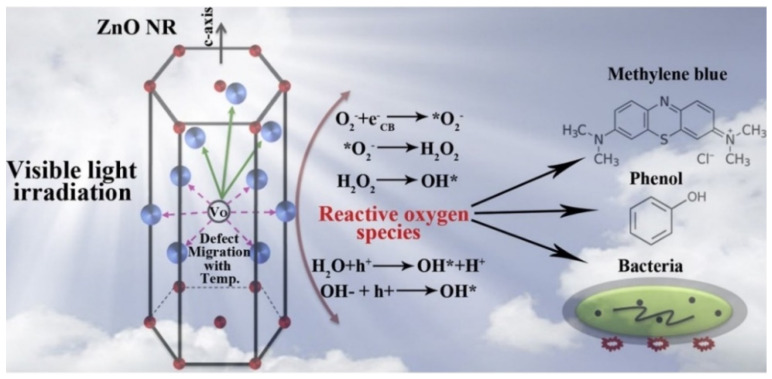
Mechanism of antibacterial and photocatalytic activity of ZnO nanorods. Adapted with permission from [43]. Copyright 2017, Elsevier.

**Figure 7 nanomaterials-12-00486-f007:**
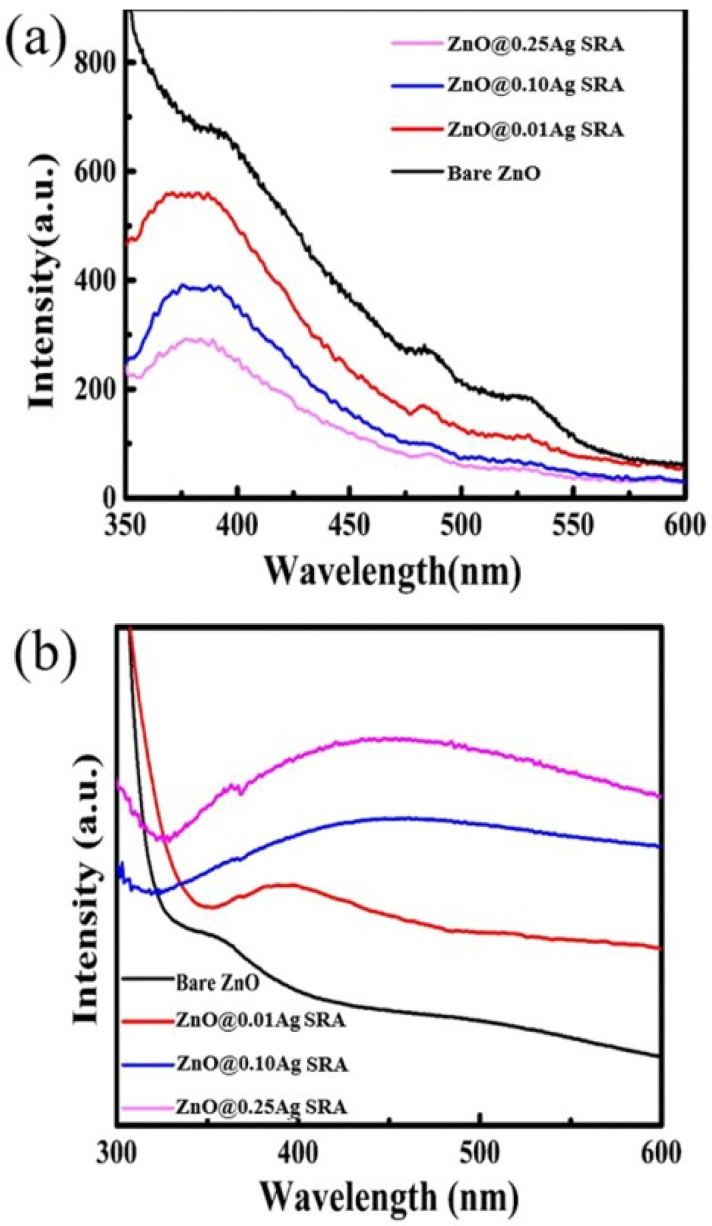
(**a**) Room temperature PL spectra of samples synthesized by electrolytes containing different concentrations of Ag^+^. (**b**) UV/Vis absorption spectra of bare ZnO and Ag-ZnO submicrometer rods. Adapted with permission from [49]. Copyright 2019, ACS.

**Figure 8 nanomaterials-12-00486-f008:**
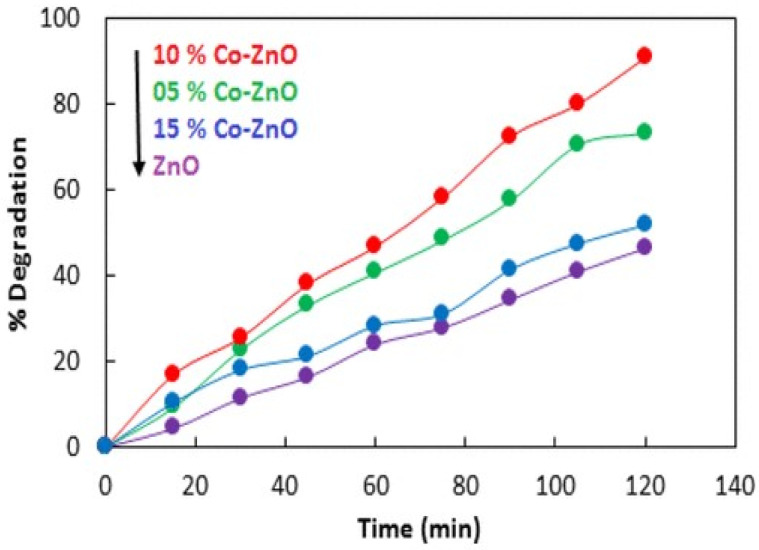
Photodegradation of methyl orange with pure ZnO and doped ZnO with 5% Co, 10% Co, and 15% Co. Adapted with permission from [56]. Copyright 2019, ACS.

**Figure 9 nanomaterials-12-00486-f009:**
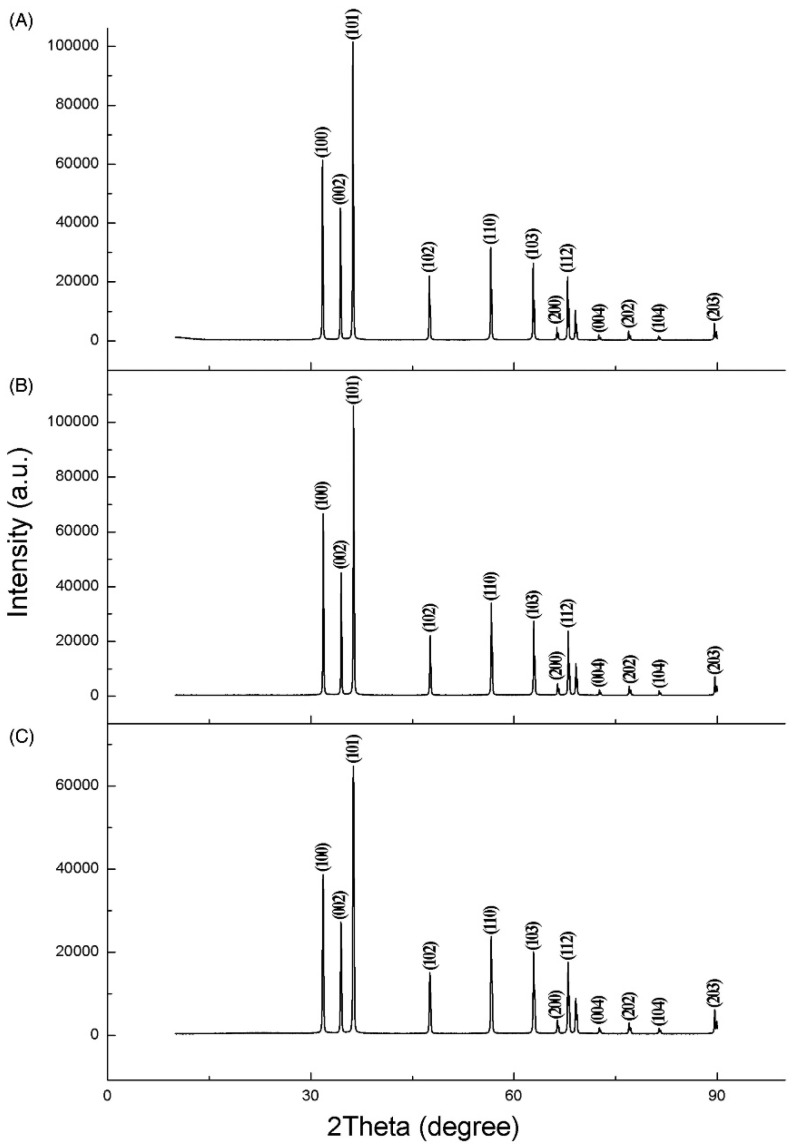
XRD spectra of synthesized zinc oxide nanoparticles using (**A**) olive leaves, (**B**) chamomile flower, and (**C**) red tomato fruit [88].

**Table 1 nanomaterials-12-00486-t001:** Percentage degradation of various dyes using doped ZnO photocatalysts.

Sr. No.	Doped ZnO	Pollutants	% Degradation	Light Source	References
1.	Sn/ZnO	Methyl blue dye	81	150 W Xe lamp	[68]
2.	La/ZnO	Methyl orange dye	85.86	Visible light	[69]
3.	Ir/ZnO	Malachite green	>90	9 W fluorescent visible lamp	[70]
4.	Sr/ZnO	Methylene blue	50	500 W Xe lamp	[71]
5.	V/ZnO	Malachite green	>99	Osram Lumilux daylight lamp	[72]
6.	Co/ZnO	Methylene blue	98	% 500 W halogen lamp	[73]
7.	Cu/ZnO	Direct blue 15 dye	70	Visible light	[74]
8.	Ag/ZnO	Cibacron brilliant yellow 3G-B	65	400 mW·cm^−2^ Xe lamp	[75]
9.	Al/ZnO	Naphthol green B	100	25 W·cm^−2^ sunlight	[76]
10.	Nd/ZnO	Congo red	93.7	Visible light	[77]

## Data Availability

Article was written in the light of all mentioned references.

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
