# Peer review of "Highly Efficient Visible Light Active Doped ZnO Photocatalysts for the Treatment of Wastewater Contaminated with Dyes and Pathogens of Emerging Concern"

_nanomaterials, 2022, doi:10.3390/nano12030486_

Round 1
Reviewer 1 Report
This manuscript focuses on the literature review of doped ZnO photocatalysts for the treatment of wastewater contaminated with dyes and pathogens of emerging concern. The review is not that novel and lacks proper justification. This manuscript needs major revision and I have listed these issues and recommendations in chronological order. Following is a summary of the major corrections and revisions:
- The authors may need to briefly address the difference(s) between the current manuscript and other similar published review articles in the Introduction section.
- In general, the references in the introduction are poorly chosen, when compared to the sentences they serve as confirmation for.
- Please explain why you have restricted the application for dyes and pathogens, over other emerging pollutants such pharmaceutical compounds and pesticides, for example? Please comment.
- The authors should make a table regarding the several doped ZnO photocatalysts and the target pollutant. They also should mention the % of degradation for each pollutant and the light source that was used for the photocatalysis method (and what is the intensity of the light source).
- Can these methods be scaled up? Cite updated papers in the said query, include it in the introduction, and conclusion part of your revised Manuscript.
- Conclusions need to be improved by specifying the discussed important points within this work. In the conclusions, the authors should also provide an outlook of the challenges and potential future directions.
Reviewer 2 Report
In this paper, the author makes a series of discussions on water pollution, focusing on the degradation of water pollutants by zinc oxide and its complexes. The discussion of the article is more specific and the introduction is very detailed. The author needs to solve the following problems before publishing.
- Figure 1 is not clear enough, and the author needs to submit clear pictures. In addition, the author needs to note the literature source of the picture.
- The author needs to list the work done by researchers in recent years on the regulation of ZnO energy band.
- As for ZnO photocatalysts, some related literatures need to be mentioned by the authors, such as: Blue Luminescent ZnO Nanoclusters Stabilized by Esterifiable Polyamidoamine Dendrimers and their UV-Shielding Applications; Palladium-catalyzed double coupling reaction of terminal alkynes with isocyanides: a direct approach to symmetricalN-aryl dialkynylimines; Preparation and Photoelectrocatalytic Performance of Fe2O3/ZnO Composite Electrode Loading on Conductive Glass; Blue Luminescent ZnO Nanoclusters Stabilized by Esterifiable Polyamidoamine Dendrimers and their UV-Shielding Applications; A near infrared fluorescence imprinted sensor based on zinc oxide nanorods for rapid determination of ketoprofen.
Round 2
Reviewer 1 Report
Authors have revised the manuscript according the recommendations, and answered the questioned points. Now it looks suitable for publication in Nanomaterials.